# Fuzzy Cognitive Maps as a Tool for Identifying Cognitive Conflicts That Hinder the Adoption of Healthy Habits

**DOI:** 10.3390/ijerph19031411

**Published:** 2022-01-27

**Authors:** Luis Angel Saúl, Alejandro Sanfeliciano, Luis Botella, Rafael Perea, Jose Antonio Gonzalez-Puerto

**Affiliations:** 1Faculty of Psychology, Universidad Nacional de Educación a Distancia (UNED), 28040 Madrid, Spain or asanfelic1@alumno.uned.es (A.S.); jrplcorreo@gmail.com (R.P.); jagonzalez@psi.uned.es (J.A.G.-P.); 2Facultat de Psicologia, Ciències de l’Educació i de l’Esport Blanquerna, Universitat Ramon Llull, 08022 Barcelona, Spain; lluisbg@blanquerna.url.edu

**Keywords:** health promotion, healthy habits, adopting healthy behaviour, difficulties or barriers, fuzzy cognitive maps, cognitive conflicts, constructivism, psychology of personal constructs

## Abstract

Implementing healthy lifestyle habits can take a great effort and sticking to such prescriptions is complicated. Failure rates amongst people seeking to adopt a healthier diet are estimated to be around 80%. Exploring the network of meanings that an individual associates with adopting habits such as healthy eating, maintaining the correct weight, and practising physical exercise can reveal the inconsistencies, obstacles, or psychological conflicts that hinder change and target-achievement. Fuzzy cognitive maps (FCM) can be of great utility in this task as they allow us to explore the structure of the personal meaning system of an individual as well as determine any obstacles and simulate hypothetical scenarios that project its future evolution. This can help to identify the foci of cognitive conflicts that hinder the adoption of healthy habits and establish more effective personalised intervention programmes that make it easier to maintain these habits.

## 1. Introduction

In 2004, as part of its policies on disease prevention, the World Health Organization approved a global strategy on diet, physical activity, and health [1], which sought to present the best available evidence on the influence of diet and physical activity on disease. It concluded that improving the adoption and maintenance of healthy lifestyle habits has a more significant impact on the health of the population than any medical treatment. Similarly, Goal 3 of the Sustainable Development Goals (2015–2030) promoted by the United Nations and approved by all 193 member states on 25 September 2015 is Healthy Lives.

As the Organisation for Economic Co-operation and Development reports, the need for health promotion and prevention can be measured in terms of its socio-economic effect [2]. Indeed, such is its impact that any investment in preventing obesity has up to a six-fold economic return.

However, adopting and maintaining healthy habits takes a great effort and adherence to this type of regimen stands at around 50% [3]. According to the Spanish Society for the Study of Obesity [4], 81% of people embarking on a diet fail in the attempt.

The connection between the self-beliefs of an individual and the meanings related to healthy eating and exercise, and his/her adherence to healthy habits has been empirically proven. Stronger self-views as a healthy eater have been found to be a significant predictor of stronger healthy eating intentions [5] and also a predictor of dietary intake [6]; the self-as-doer identity predicted improved healthy food consumption [7]. Likewise, self-schemas and identities have been related to adherence to exercise [8,9] with transition into exercise involving a complex and dynamic process of identity transformation [10] and personal meanings associated with long-term adherence [11]. In a few cases, such psychological factors hinder the adoption of healthy habits and the achievement and maintenance of a reduced body weight [12,13]. In most cases, merely wishing to lose weight has no effect. Subjects themselves sometimes sabotage—not entirely consciously—their own desired actions in following a diet or increasing their physical activity. However, in addition to the purely volitional aspects (eating less and getting more exercise), there are other physiological, genetic, and psychological issues that significantly condition the way a person may or may not manage to fulfil their desire to reduce their body weight. At the same time, our contemporary environment with its constant stimuli and temptations makes it very difficult to achieve such a simplified solution.

The constructs around what it means to maintain healthy lifestyle habits and the psychological implications of following health advice are fundamental in determining whether such indications of incorporating healthy habits, diet, and exercise are successful, providing that there are no other metabolic complications preventing this outcome. It is, therefore, essential to identify the psychological obstacles (specific psychological variables or cognitive conflicts) that prevent or hinder the desired process of a person and the psychological variables that facilitate that process in order to create intervention programmes that offer the highest possible level of efficiency.

From the perspective of personal construct psychology (PCP) [14], it is possible to assess the structure and content of the personal meanings each person employs and how these may interfere with the fulfilment of their goals. Psychological assessment using the repertory grid (RG) allows us to identify cognitive conflicts between the aspects that the individual wants to change and those that he/she wants or needs to maintain [15]. The findings of our research team over the last 20 years (GICUNED, https://blogs.uned.es/gicuned/en/home/ accessed on 22 December 2021) show how this type of conflict impacts on the adoption of desired behaviours, relating the presence of these conflicts to the non-attainment of the intended goals of an individual [16,17,18]. Using the latest innovations and software developed by our group for evaluating the structure of the personal construct system (PCS) and for representing fuzzy cognitive maps (FCMs) of a subject [19], we can create digraphs of that structure as well as graphs showing the dynamics by which that PCS evolves. These developments allow us to test and simulate their anticipated dynamics in hypothetical scenarios of change and to gain a fuller understanding of their systemic properties. This is of great help in mathematising these psychological processes and creating personalised reports of the strengths and difficulties in achieving the desired goals of subject, representing scenarios of evolution from their existing situation or different scenarios of change if certain constructs are modified. Apart from the interest of offering the scientific community tools that can facilitate the exploration and evaluation of psychological change processes, this type of feedback gives subjects themselves the power of change and facilitates motivational accompaniment by the healthcare staff involved.

This article demonstrates that FCMs can be a very useful tool in identifying the cognitive conflicts that hinder the adoption of healthy habits even if these are desired by the subject. This paper sets out the methodology used for creating FCMs from an exploration of the PCS of an individual as well as the supporting mathematics. A simulated case is presented as an example of the procedure.

### Humans as Constructors of Reality

In order to understand the way humans interact with the world, we began from a constructivist position whereby each individual gives meaning to the world through personal constructs. The most fully elaborated theory in this regard is PCP, developed by George Kelly [14]. Kelly structured the entire construction system of an individual through personal constructs. A personal construct is the basic unit of construction. It is a bipolar dimension of meaning that individuals use to make sense of, order, and predict their experience. For example, using the “good person vs. bad person” construct, a person can sort different elements (people in this case) into the two poles of that dimension. They can even place themselves (Self-Now), their ideal (Ideal-Self), etc., in these poles, thus defining their identity. In this sense, a construct in which both the Self-Now of a subject and their Ideal-Self are at the same pole indicates that they are content and do not wish to change. This is what we call a congruent construct. On the other hand, if the subject places him or herself at one pole of a construct (Self-Now) but would like to be at the opposite pole (Ideal-Self), this indicates dissatisfaction and a desire for change. This is what is known as a discrepant construct.

The personal construct system (PCS) of an individual is finite; the constructs are arranged in a hierarchical and related structure. In 1955, Kelly proposed the RG as a tool for exploring the PCS of an individual (for a review of the technique, see Fransella [20]). In later developments, Hinkle [21] proposed another related technique, the Implications Grid (ImpGrid), for exploring the implications of this system. Identifying the repertoire of personal constructs that an individual uses and understanding their structure and dynamics helps to understand the position from which they interact with the world. The dynamics of the PCS are not perfect nor are they free from difficulties and inconsistencies; there may be conflicts that hinder the flow and, sometimes, the achievement of desires and needs for change of the subject [16].

Therefore, the ability to represent the structure of a PCS of an individual, determine the implications and relationships between certain constructs and others, and record their cognitive inconsistencies or conflicts would be of great help in identifying where the difficulties of the system lie in achieving the desired goals of adopting healthy habits. This paper proposes the use of FCMs for this purpose [19].

## 2. What Is an FCM?

FCMs were first proposed by Kosko [22] to enable the representation of mental models formulated in a natural language through the use of fuzzy logic. This technique allows us to see and simulate future scenarios based on the way the flow of causality is distributed. We wanted to apply this technique to the PCS of an individual, specifically in the area of promoting healthy habits. In this way, starting with an exploration of the meaning system of an individual and his or her network of causal attributions, our aim was to detect inconsistencies in the system and explore scenarios for change. To conduct this interpretation, we used two means of representing FCMs: map digraphs (MDs) and graphs of PCS dynamics.

### 2.1. Parts of an FCM

FCMs have two main components: (a) nodes or vertices, elements within the system with an associated value or activation that establishes the state at any given time; and (b) edges, connections between the elements that represent the channels along which activation is propagated. An FCM is, therefore, a network comprising nodes that transmit their activation to other nodes along edges.

In this case, we wanted to represent the PCS of a person using FCMs. We achieved this by establishing the different personal constructs of an individual as nodes in the network and giving them a degree of activation corresponding with their Self-Now perception. We then only needed to join these nodes by way of the edges; for this purpose, we used the causal attributions between the constructs (extracted from an ImpGrid). The result was an FCM showing a network of causal attributions that allowed us to study the PCS of an individual and perform simulations in hypothetical scenarios.

To graph this FCM, we drew an MD where the nodes were represented as spheres and the edges as arrows connecting those spheres. We also used additional coding (colours, sizes, thicknesses, etc.; see Figure 1) to provide a rapid view of the unique characteristics of an analysis from the PCS. Looking at the nodes in the figure, they have a name, colour, and size. The name shows the pole with which the subjects define themselves. The colour represents the orientation with respect to the ideal of that pole (red for discrepant, green for congruent, and yellow for dilemmatic). The size shows the degree of activation of that node (the larger the node, the greater the self-identification around that pole). The edges are in different colours and sizes; the size indicates the strength of the relationship (the larger the edge, the greater the involvement) and the colour indicates the type of relationship (black for a direct relationship, i.e., an increase in the first node activates an increase in the second one; red for an inverse relationship, i.e., an increase in the first node activates a decrease in the second one).

### 2.2. Mathematics Underlying an FCM

As previously mentioned, these components (nodes and edges) were quantified and structured through a mathematical model. This model was defined through: (a) an activation vector; (b) a weight matrix; (c) a propagation function; and (d) a threshold function. The activation vector was a set of ordered numbers that defined the initial state of the system, i.e., the initial activation of each of the nodes. The weight matrix was a matrix whose elements were the intensity and direction of the causality relations between the nodes of the system, i.e., the quantification of the edges. The propagation function was the mathematical equation that related the nodes and edges to establish how the activation was propagated through the network. Finally, the threshold function defined the degree of activation of a node according to the activation coming from the rest of the nodes, ensuring it remained within a suitable margin. For a more complete explanation of the mathematical foundations of FCMs, see Kosko [22], Stylios and Groumpos [23,24], and Tsadiras [25].

When computing the FCM applied to the PCS of an individual, we used the vector of the Self-Now extracted from the RG as the activation vector. Thus, the activation of the nodes represented the self-perception of the subjects evaluated. For the edges, we then used the causal attributions obtained from the ImpGrid as a weight matrix. In this way, we created an FCM that enabled us to study the PCS of the subject and simulate possible evolutions in the self-perception of the subject according to their causal attributions. In the example below, we present the type of RG and ImpGrid proposed.

To summarise, by combining FCMs with personal construct theory it was possible to create maps of the PCS of an individual with the following characteristics [19]:They represented images of the causal attributions of the person constructing the map.They used intrinsically fuzzy linguistic elements (they did not belong to an absolute all-or-nothing meaning).They enabled hypothetical scenarios to be simulated and the dynamics of specific situations to be studied through activation vectors.

### 2.3. Creation of an FCM from the RG

The specific methodology for applying an FCM using the RG can be seen in the flowchart in Figure 2. Here, it can be seen that the starting point was to apply and obtain the RG of the subject being assessed. This helped us to obtain the personal constructs of the subject together with their scoring matrix. From the latter, we were interested in extracting two specific vectors: the Self-Now and the Ideal-Self. Using the information from the Self-Now vector and with the personal constructs of the individual, we could then apply the ImpGrid to obtain the weight matrix. Having obtained this matrix, all that remained was to create an initial activation vector using the information contained in the Self-Now in order to infer scenarios. This inference gave an iteration matrix that showed a simulation of how the Self-Now evolved. This matrix, combined with the Ideal-Self vector, allowed us to plot a first graph of the PCS dynamics, explained below. To draw the MD, we only had to choose one of these scenarios from the iteration matrix and express it as a network. We used a variety of open-source software platforms to perform these steps [26,27,28,29].

## 3. Application of FCMs in the Promotion of Healthy Habits

To explore the construction system in the field of promoting healthy habits, we proposed the following version of the RG, which comprised 12 constructs and 11 elements (see Appendix A). We proposed six predefined constructs related to the construction of the subjects with regard to their weight, sport, food, and health; we presented questions to elicit six other constructs from their PCS (see Appendix B).

Among the elements to be evaluated, we used the Self-Now as well as a one-year projection of the self and the Ideal-Self, three significant people in their life (father, mother, and partner or other significant person), the roles related to people who are and are not overweight, people who have and do not have healthy habits, and the role of *persona non grata.*

To score the RG, the subjects had to determine at which pole (left or right) they placed each of the elements for each construct using the following scale: for the left pole, 1 (“very” like that pole), 2 (“quite”), and 3 (“slightly”); for the right pole, 5 (“very” like that pole), 6 (“quite”), and 7 (“slightly”). A score of 4 could be given when the element could not be placed at one pole or the other.

We present an example of a simulated grid in Figure 3 that we will use to illustrate the representations of an FCM and explore its usefulness.

To understand how these scores were interpreted, let us review the construct in the first row, overweight vs. correct weight. In this construct, the subject rated himself as “very” overwight (score of 1); in a year’s time he did not know whether he would see himself as overweight or at the correct weight (4); he viewed his father as “quite” overweight (2), his mother as “slightly” at the correct weight (5), his partner as “slightly” overweight (3), etc. His ideal in this construct placed him as “very” at the correct weight (7).

In the following construct, he perceived himself as “slightly” uncontrolled in eating and his ideal would be to be “very” in the desire to control eating.

To explore the causal attribution network in relation to the identified constructs, we used the ImpGrid [21] to ask the subject to state what the implications would be of a change from their current position within a construct to the other pole. They had to indicate which other constructs would change and towards which pole, scoring them on a scale of −3 to 3. In our example, for this subject to go from the fat pole to the correct weight pole would also involve a change in three other constructs (see Figure 4). It would lead him towards “very” healthy, “very” active, and it would “quite” increase his confidence.

Using all this information gathered from the first grid and the ImpGrid, we could, inter alia, give a 3D representation of the structure of the network of meanings and the causal attributions between the constructs studied.

In a two-dimensional representation of the Self-Now, most of the nodes could be seen to be red, indicating that the subject was not content with them and wanted to make changes to them (see Figure 5a). The green nodes indicated the constructs in which he felt content. The edges showed the relations between the nodes. The information presented in the MD reflected the current structure of the PCS. It gave us a view of the Self-Now PCS of the subject at the present moment, a sort of snapshot in time. This was the type of information that the RG alone could provide. However, by entering the information gathered in the ImpGrid, we could simulate the evolution of this PCS based on the network of causal attributions.

This simulation of the predicted evolution of the PCS of a subject was shown in the PCS dynamics. On the x-axis we had the time (from the present, at the origin, towards the future, in the number of iterations) and on the y-axis, the distance between the Self-Now and the Ideal-Self (i.e., the closer to the x-axis, the closer the subject was to his ideal). By only simulating approximately 30 iterations (see Figure 5b), we could see that this was a very stable system in which the desired changes did not occur. The constructs in which the subject wanted to change remained far from the ideal, i.e., at the top of the graph.

As can be seen, after a series of iterations the changes in the system were minimal. They included the definition of the healthy vs. the unhealthy construct, in which the subject in his initial state did not know where to define himself. Over time, he came to identify himself with the unhealthy pole (see Figure 6). However, we did not find any other green nodes that would indicate the achievement of a desired change.

The exploration of the relations between the poles in which the Ideal-Self was situated gave us clues to the inconsistencies in the system and the difficulty of achieving a few of the goals (see Figure 7). It should be remembered that the subject was content in the enjoys eating pole and did not want to make changes to it but his desire to adopt healthy eating and a healthy life to have control of eating or to develop his willpower were inversely related to that pole, as shown by the red arrows pointing to it. 

This information gave us clues to the difficulties he would have in achieving these goals if there was no change in his attributions with regard to the meaning he gave to enjoys eating.

The real potential of this new FCM-based methodology and its ability to simulate the evolution of the PCS of a subject was that it enabled us to test different alternatives for change and predict their most likely impact on the achievement of the desired goals, thus allowing us to select the therapeutic approach that would work best for each particular subject. For example, we could propose a simple change in behaviour so that the subject would achieve his desire to do sport and have a healthy diet without modifying any of the implications of his system. With this approach to problem-solving we could see that although he would initially achieve his goals, over time the system would return to a state that was far from his ideal (see Figure 8).

In order to achieve the desired goals, there must be a change in the implications and attributions of the enjoy eating meaning so that he could maintain his position of enjoying eating, which he was content with and did not want to change. At the same time, from this position, the ability to enjoy eating must be included—perhaps in a much more conscious way; perhaps through therapeutic interventions to enhance mindfulness—so that he could fully experience the act of eating and savouring food [30,31]. In this way, healthy food and control in eating could be related to the enjoyment of conscious eating, breaking the previous inverse relationship between the two (see Figure 9).

As we could see in the PCS dynamics (see Figure 10), if these changes in these implications occurred, the system as a whole moved towards the achievement of the wishes of the subject. We observed how practically all of the constructs moved towards the lower part of the graph and towards the desired changes identified with the Ideal-Self.

## 4. Conclusions

FCMs, with their MD and PCS dynamic representations, are very useful tools for exploring the structure of and inconsistencies in the PCS of an individual and for identifying those that hinder change and the achievement of the desires for change. This can be very useful in any field of research in which it is relevant to know how the construction systems of a subject determine their present and future behaviour. Once the structure and dynamics of the PCS have been determined, the possibility of predicting change is very valuable for determining how subjects will evolve and where they will and will not have problems in relation to cognitive conflicts that may even impede the achievement of their desire for change.

Within the field of the adoption of healthy habits and through the simulated case we have presented, we can see the value of this methodology in anticipating difficulties in adopting habits, even those that are desired by the subject. Identifying the cognitive conflicts found in the PCS is of great help both for subjects, through the feedback they can receive, and for the healthcare staff accompanying them in the process of adhering to healthy habits—in the case presented, increased physical exercise, controlled dietary intake, and a healthy diet. Although the process of adopting healthy habits is complex and depends on a wide variety of genetic, social, economic, and other variables, identifying cognitive conflicts that hinder change could help to generate foci for psychological intervention, making it easier to rework and reconstruct several of the meanings associated with the identity of a subject and their relationship with the habits being addressed. This would help reduce dropout rates. An increased adherence will undoubtedly benefit both the physical and psychological health of the person.

We consider this to be a very promising line of research with implications for interventions to improve adherence to any treatment. However, we are still at an initial phase in the development of the analysis methodology. As a possible line of work, we are considering studies to verify the utility of these tools and the fit between the simulations and the real change. In any case, we have proposed an innovative methodology for the study of subjectivity and identity construction as well as the representation and mathematisation of the structure and the dynamics of the construction systems of individuals and the study of change and the understanding of the consistency of non-change.

Determining the most effective components of intervention will help in the development of health-behaviour intervention programmes in adults that can be applied to large groups and in mass campaigns using the facilities of information and communication technologies.

Regarding the main limitations of our study, it has already been stated that its goals were to discuss the usefulness of FCMs to explore the structure of the network of personal meanings of an individual that he/she associates with adopting healthy habits to reveal the inconsistencies, obstacles, or psychological conflicts that hinder change and target-achievement. We have exemplified such usefulness with a detailed description of a simulated case study; thus, the limitations are the classical ones in simulation studies: (a) they do not yield a simple and single answer but provide a set of system responses to different operating conditions; (b) they are approximative methods that may produce different solutions in repeated runs; and (c) the difficulty in finding the optimal values increases due to an increase in the number of parameters. In the case of the topic of our study, however, these limitations were intrinsically intertwined with the complexity of the very phenomenon being studied.

## Figures and Tables

**Figure 1 ijerph-19-01411-f001:**
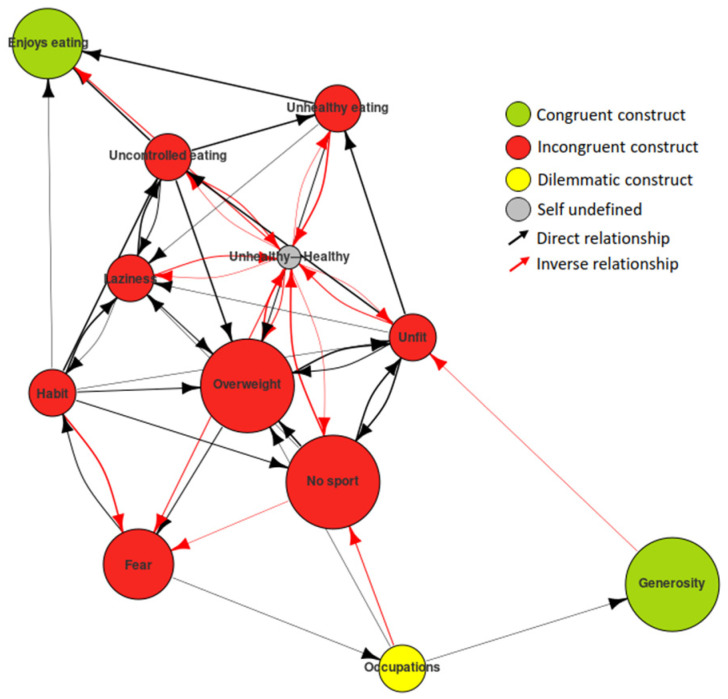
FCM coding.

**Figure 2 ijerph-19-01411-f002:**
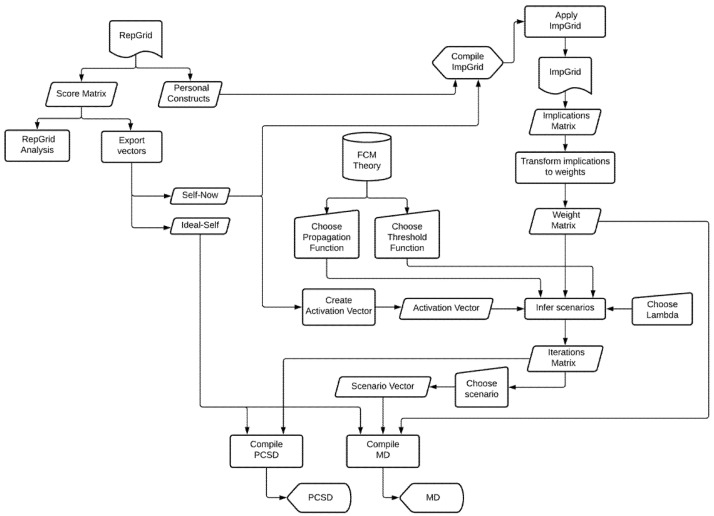
Creation of an FCM from an RG.

**Figure 3 ijerph-19-01411-f003:**
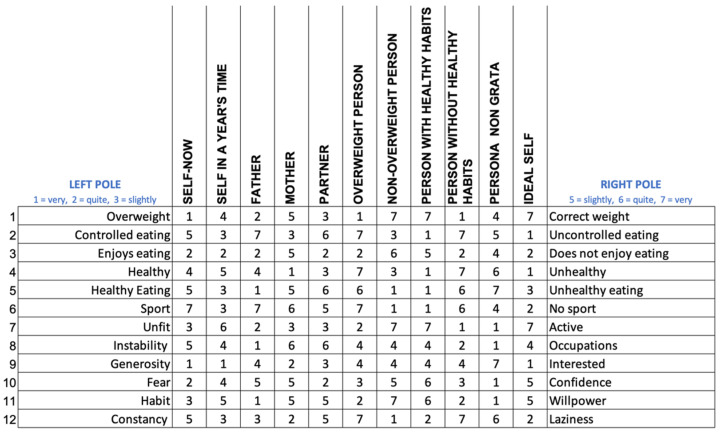
Repertory grid.

**Figure 4 ijerph-19-01411-f004:**
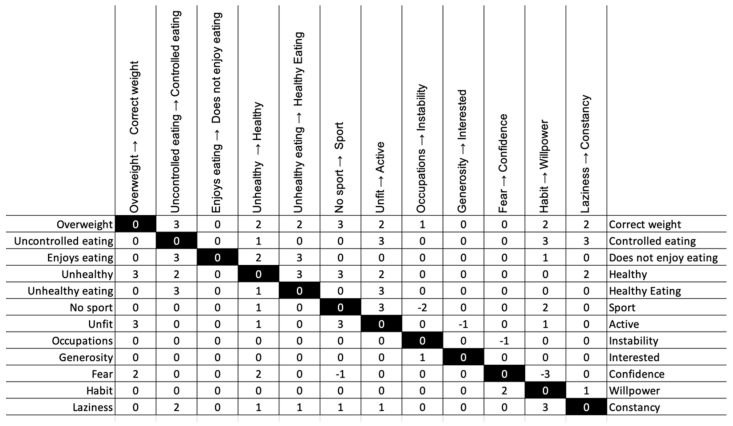
Implications grid: if you go to the other pole, what other constructs would change?

**Figure 5 ijerph-19-01411-f005:**
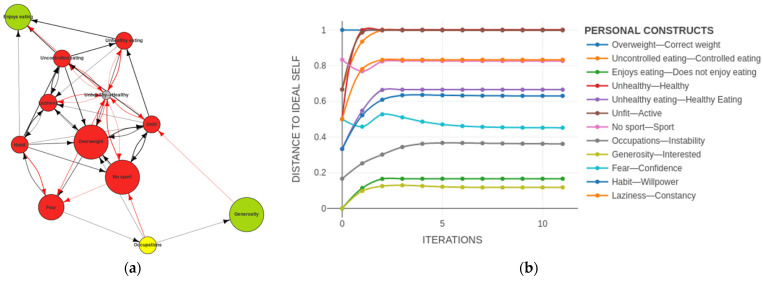
(**a**) Self-Now map digraph: self initial state; (**b**) personal construct system dynamics: Self-Now dynamics after 30 iterations.

**Figure 6 ijerph-19-01411-f006:**
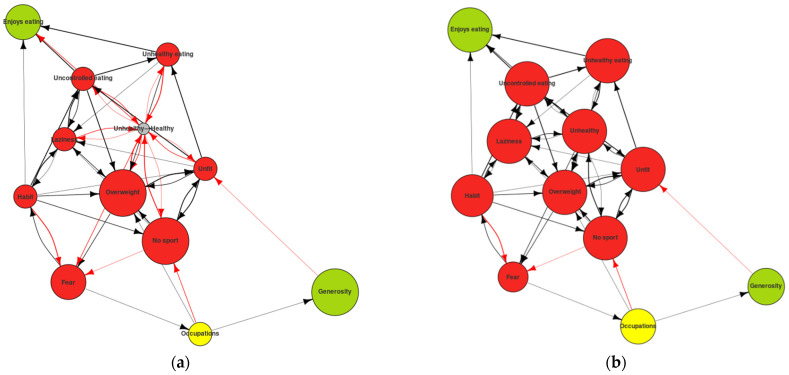
Self-Now map digraph: (**a**) Self-Now initial state; (**b**) Self-Now simulation after 11 iterations.

**Figure 7 ijerph-19-01411-f007:**
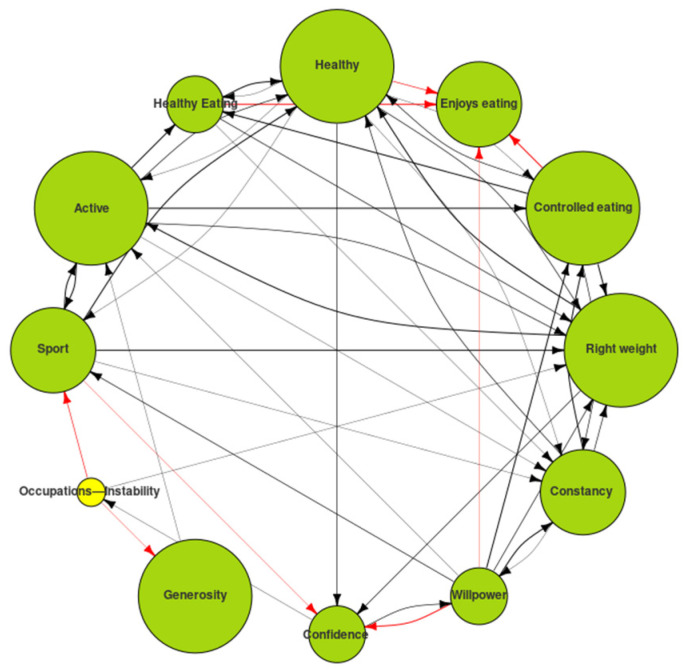
Ideal map digraph: exploring inconsistencies in the system.

**Figure 8 ijerph-19-01411-f008:**
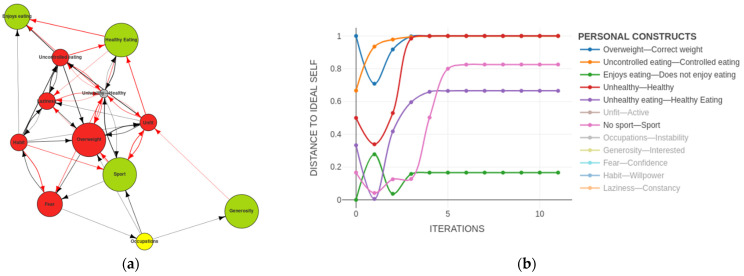
Simulation of Self-Now doing sports and healthy eating: (**a**) Self-Now map digraph initial state; (**b**) personal construct system dynamics.

**Figure 9 ijerph-19-01411-f009:**
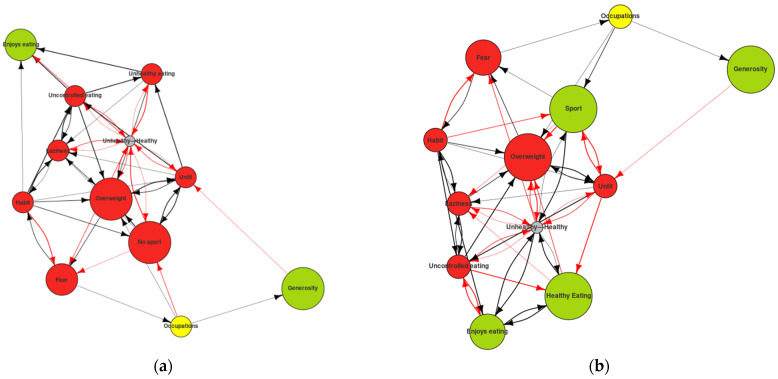
Simulation of change in the implications: (**a**) Self-Now map digraph initial state; (**b**) Self-Now map digraph with change in the implications.

**Figure 10 ijerph-19-01411-f010:**
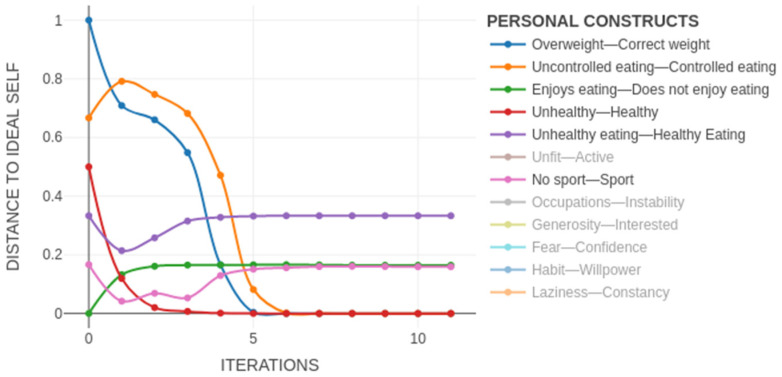
Personal construct system dynamics of the simulation change in the implications.

## Data Availability

Data are contained within the article.

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
