# Peer review of "Fuzzy Cognitive Maps as a Tool for Identifying Cognitive Conflicts That Hinder the Adoption of Healthy Habits"

_ijerph, 2022, doi:10.3390/ijerph19031411_

Round 1

Reviewer 1 Report

Some revisions for the authors to consider:

1. It would be better to add limitation and implication in details.
2. Please check abbreviations. For example, when authors use RGT first, please spell out the full term. Additionally, does PERS mean personal in the Figures?
3. Regarding references not in English, this may be followed by an English translation of the title in square brackets. Please check the journal's guideline. 
4. It would be important to clarify terms/definitions in the Grids, such as right weight, overweight, and uncontrolled/controlled.

Author Response

  1. It would be better to add limitation and implication in details.

Thank you very much for your comments for improvement. In the conclusions section, we have added the following paragraph (lines 401-413):

“Regarding the main limitations of our study, it has already been stated that its goals are to discuss the usefulness of FCMs so as to explore the structure of an individual's network of personal meanings that he/she associate with adopting healthy habits and reveal the inconsistencies, obstacles, or psychological conflicts that hinder change and target-achievement. We have exemplified such usefulness with a detailed description of a simulated case study, and thus the limitations are the classical ones in simulation studies: (a) they do not yield a simple and single answer but provide a set of the system’s responses to different operating conditions; (b) they are approximative methods that may produce different solutions in repeated runs, and (c) the difficulty in finding the optimal values increases due to an increase in the number of parameters. In the case of the topic of our study, however, these “limitations” are intrinsically intertwined with the complexity of the very phenomenon being studied.”

  1. Please check abbreviations. For example, when authors use RGT first, please spell out the full term. Additionally, does PERS mean personal in the Figures?

We deleted the abbreviations that appeared only once because they were unnecessary. We made sure that abbreviations appear after the first appearance of the term and then only the abbreviation follows.

PERS. in the Figure 3 and appendix A means person. We have changed these two figures.

  1. Regarding references not in English, this may be followed by an English translation of the title in square brackets. Please check the journal's guideline. 

Thank you for pointing out this error. We have translated all the references that appeared in Spanish into English, inserting them between square brackets, as can be seen in the references section.

  1. It would be important to clarify terms/definitions in the Grids, such as right weight, overweight, and uncontrolled/controlled.

These “terms” are in fact constructs suggested by the authors but their specific meanings depend on the participants personal use of them (that’s why they are personal) this personal nature is an intrinsic part of the way the data are analyzed and also of Personal Construct Theory and Fuzzy Logic.

Reviewer 2 Report

Thank you, this was an interesting paper that provides a useful tool for clinician and health experts to adopt.

In general, I liked the paper. It was well written and easy to understand. What was missing for me in the introduction was whether there is research that shows that when participants rework their meaning structure associated with their identity and their relationship with their eating habits that there are, in fact, lower drop-out rates and more health success. In other words, is cognitive-rerouting the solution to the problem? If the authors can establish this point in the introduction, I think their tool is more convincing.

The other thing that was a bit difficult while reading the paper were the abbreviations. I think one or two abbreviations make sense to increase the reading flow. But with the current manuscript, I kept jumping back and forth to find the correct description of the abbreviation I was currently reading (which might also correlate with my terrible memory). In addition, introducing ICT as abbreviation in the last sentence as the last word, for example, fulfills little purpose.

Here are some other minor questions I had regarding the paper:

  • I didn’t understand what a direct relationship in the Fuzzy Cognitive Map Coding was compared to the inverse relationship. Is one positive and the other negative?
  • How did the authors choose the six predefined constructs?
  • Is the demonstration in Figure 5b and 8b just a hypothetical demonstration of how these constructs stabilize across time or is it based on longitudinal data?

Author Response

Thank you, this was an interesting paper that provides a useful tool for clinician and health experts to adopt.

In general, I liked the paper. It was well written and easy to understand. What was missing for me in the introduction was whether there is research that shows that when participants rework their meaning structure associated with their identity and their relationship with their eating habits that there are, in fact, lower drop-out rates and more health success. In other words, is cognitive-rerouting the solution to the problem? If the authors can establish this point in the introduction, I think their tool is more convincing.

Thank you very much for your very enriching comments. In the introduction we have added studies on the connection between self-beliefs and the individual's meanings related to healthy eating and exercise and their adherence to these healthy habits. We added the following information (lines 38-57): The connection between (a) the individual’s self-beliefs and meanings related to healthy eating and exercise and (b) his/her adherence to healthy habits has been empirically proven. Stronger self-views as a healthy eater have been found to be a significant predictor of stronger healthy eating intentions (Malek et al, 2017), and also a predictor of dietary intake (Noureddine & Stein, 2007); the self-as-doer identity predicted improved healthy food consumption (Brouwer & Mosack, 2015). Likewise, self-schemas and identities have been related to adherence to exercise (Cooke, 2020; Strachan & Whaley, 2013), with transition into exercise involving a complex and dynamic process of identity transformation (Mcgannon, 2002), and personal meanings associated with long-term adherence (Springer, 2005).

The other thing that was a bit difficult while reading the paper were the abbreviations. I think one or two abbreviations make sense to increase the reading flow. But with the current manuscript, I kept jumping back and forth to find the correct description of the abbreviation I was currently reading (which might also correlate with my terrible memory). In addition, introducing ICT as abbreviation in the last sentence as the last word, for example, fulfills little purpose.

We deleted the abbreviations that appeared only once because they were unnecessary. We made sure that abbreviations appear after the first appearance of the term and then only the abbreviation follows.

Here are some other minor questions I had regarding the paper:

  • I didn’t understand what a direct relationship in the Fuzzy Cognitive Map Coding was compared to the inverse relationship. Is one positive and the other negative?

Thanks for your comment.  A direct relationship (+) entails that an increase in the first node activates an increase in the second one.An inverse relationship (-) entails that an increase in the first node activates a decrease in the second one.

We have introduced this explanation between lines 178 and 180.

  • How did the authors choose the six predefined constructs?

We choose these six predefined constructs because they appear repeatedly and consistently in our own clinical practice when working with clients with difficulties in following a healthy diet.

  • Is the demonstration in Figure 5b and 8b just a hypothetical demonstration of how these constructs stabilize across time or is it based on longitudinal data?

It is a projection of the dynamics of the system along time. So, it is a simulation of how the system will evolve according to its initial state.

Round 2

Reviewer 2 Report

Thank you for addressing my comments!